# Impact of Cross-Linked Hyaluronic Acid on Osteogenic Differentiation of SAOS-2 Cells in an Air-Lift Model

**DOI:** 10.3390/ma15196528

**Published:** 2022-09-20

**Authors:** Bianca Nobis, Thomas Ostermann, Julian Weiler, Thomas Dittmar, Anton Friedmann

**Affiliations:** 1Department of Periodontology, School of Dentistry, Faculty of Health, Witten-Herdecke University, Alfred-Herrhausen-Str. 50, 58455 Witten, Germany; 2Institute of Immunology, Centre of Biomedical Education and Research (ZBAF), Stockumer Str. 10, 58448 Witten, Germany; 3Department of Psychology, Witten-Herdecke University, 58455 Witten, Germany

**Keywords:** cross-linked hyaluronic acid, cross-linked collagen substrate, air-lift organoid model, SAOS-2 cells, osteogenic differentiation

## Abstract

The aim of this study was to investigate the impact of cross-linked hyaluronic acid on osteoblast-like cells seeded on top of two collagen substrates, native porcine pericardium membrane (substrate A) and ribose cross-linked collagen membranes (substrate B), in an air-lift model. Substrates A or B, saturated with three hyaluronic acid concentrations, served as membranes for SAOS-2 cells seeded on top. Cultivation followed for 7 and 14 days in the air-lift model. Controls used the same substrates without hyaluronic pre-treatment. Cells were harvested, and four (Runx2, BGLAP, IBSP, Cx43) different osteogenic differentiation markers were assessed by qPCR. Triplicated experiment outcomes were statistically analyzed (ANOVA, *t*-test; SPSS). Supplementary histologic analysis confirmed the cells’ vitality. After seven days, only few markers were overexpressed on both substrates. After 14 days, targeted genes were highly expressed on substrate A. The same substrate treated with 1:100 diluted xHyA disclosed statistically significant different expression level vs. substrate B (*p* = 0.032). Time (*p* = 0.0001), experimental condition as a function of time (*p* = 0.022), and substrate (*p* = 0.028) were statistically significant factors. Histological imaging demonstrated vitality and visualized nuclei. We conclude that the impact of hyaluronic acid resulted in a higher expression profile of SAOS-2 cells on substrate A compared to substrate B in an air-lift culture after two weeks.

## 1. Introduction

The bone remodeling process is mediated by osteoblasts. To start this process, the mesenchymal stem cells are influenced by signaling molecules. These signaling molecules drive several pathways in which the upregulation of transcription factors such as Runx2 plays a crucial role in osteoblast differentiation. BGLAP (osteocalcin) is synthesized and secreted by mature osteoblasts and osteocytes in the later stages of bone formation. It is also thought to participate in bone mineralization through its ability to bind to hydroxyapatite [1,2,3,4]. IBSP, also known as osteopontin, is a glycoprotein of the extracellular matrix and is associated with mineralized tissue [5,6,7,8,9]. The upregulation of transmembrane protein Cx43 expression discloses the maturation stage of an osteoblast [10]. The extracellular matrix protein periostin (POSTN) is expressed at the late differentiation stage of an osteoblast into the col-1 matrix and accelerates cortical bone formation under mechanical load [11]. Cell cultures are a useful model for basic research regarding the regulation of bone cell interactions [12]. The principle of the air-lift model as described in the literature allows high cell density with specific cell-to-cell and cell-to-matrix contacts, therefore mimicking its best environmental conditions in vivo. While previous experiment used metal grids to lift up the cell pellet above the medium level, today, disposable inserts for 6-well plates are available for this purpose [13]. Thus, placing the cell pellet either on top of the insert’s membrane or on an additional substrate adapted in size, the nutrition progresses via diffusion, as the cells are detached from direct contact with the medium. The “organoid” model proposed by the group of Zimmermann et al. used cells from the rat calvaria and detected the onset of mineralization at the air/medium interface already after 7 days under histotypic conditions [14]. The nutrition of osteoblast-like SAOS-2 cells is arranged similarly but on several semi-permeable membrane materials was proven sufficient for up to 28 days in culture. This previous study from our group demonstrated that in serving as a membrane for cell arrangement, neither collagen nor e-PTFE membranes represented a barrier to the nutrient’s diffusion.

Modern hyaluronic acid became the focus of the current research as the presence of this molecule showed impressive impact on cells of different origin in vitro and in vivo. Hyaluronic acid is present in high concentrations mainly in soft connective tissues such as the skin, umbilical cord, synovial fluid [15], vitreous body [16], and periodontium [17]. Similarly, hyaluronic acid increased osteoblast activity by stimulating the differentiation and migration of mesenchymal stem cells [18,19,20]. Moreover, several studies showed bacteriostatic [21,22], anti-inflammatory [20], anti-edematous [17], proangiogenic [23], and osteoinductive [24,25,26] properties. Based on the in vitro data for cross-linked hyaluronic acid gel (xHyA) regarding the stimulation of osteoblasts [27], this study evaluated additional effects of hyaluronic acid on in vitro stimulation of osteoblast-like cells (SAOS-2 cells).

The previous experiment confirmed that the three questioned substrates were capable of keeping the permeability level high to nourish the cells sufficiently, as assessed by their transcription rates. In this study, one different substrate was included and using the same osteogenic additives as in the past experiment, we included the cross-linked hyaluronic acid treatment for the substrates to evaluate the potential impact on the transcription rate of the bone-like cells [28]. Following the setup from the previous study [13], the cross-linked collagen membranes served as substrates and potential carriers for xHyA, as indicated by a recent pre-clinical study [29]. Both collagen membranes chosen represented cross-linked porcine collagen devices. However, the substrate A of natural pericardium collagen (NPCM) was less cross-linked and less compressed compared to the substrate B (ribose cross-linked collagen membrane; RCCM), which undergoes the sugar cross-linking and compression by pressure for increased cell occlusivity at the end of the manufacturing process.

## 2. Materials and Methods

Human osteoblast-like Saos-2 cells (DSMZ, Braunschweig, Gemany) were grown in Dulbecco’s modified Eagle’s medium (PAN Biotech, Aidenbach, Germany) supplemented with 15% fetal bovine serum (Biochrom GmbH, Berlin, Germany), 10.000 units penicillin, and 10 mg/mL streptomycin (Merck, Darmstadt, Germany). The medium was changed every 48 to 72 h. Cells used for the air-lift setup were taken from the passages 10–19 once the confluence level was established. For osteoblast differentiation, media supplements contained 50 μg/mL ascorbic acid (Merck, Darmstadt, Germany), 10 mM β-glycerophosphate (Merck, Darmstadt, Germany), and 1% amphotericin B (Merck, Darmstadt, Germany). The negative control (NC) contained medium alone; the positive control (PC) contained the abovementioned differentiation stimuli (Figure 1).

For the air-lift experiments (Figure 2), 2 × 10^5^ SAOS-2 cells were seeded in an “organoid” order on two collagen substrates, both either previously hydrated with the xHyA formulation or used as native materials. High molecular weight hyaluronic acid formulation with ca. 1 MegaDalton (1000 kDa) was chosen in the cross-linked condition. The degree of cross-linking corresponded to the medium cross-linking rate with a degradation time of approximately 3–4 weeks according to the manufacturer (Regedent AG, Zürich, Switzerland). The cross-linking agent was BDDE (1.4-butanediol di-glycidyl-ether). The difference between the two substrates was the nature and grade of collagen cross-linking; one was cross-linked by representing porcine pericardium ((NPCM, Smartbrane^®^; substrate A) and one (substrate B) by ribose added during the patented manufacturing process RCCM, Ossix^®^Plus; Regedent AG, Zürich, Switzerland). The pre-shaped sterile substrates fitted the size of the air-lift inserts. The amount of medium pipetted into 6-well plates was low enough to keep the level of nutrients in contact with the bottom side of the substrates positioned in the inserts mounted into the wells.

While the culture medium alone accounted for the negative control group, the culture medium containing ascorbic acid and β-glycerophosphate as supplements for osteogenic differentiation was applied to the positive control group. The impact of three hyaluronic acid concentrations on adjunctive osteogenic was estimated by calculating the difference in the gene expression rate after the xHyA hydration in relation to the rate from the positive control. The difference in the expression values between the rates from the negative control and test groups showed the upregulation of the target genes under the combined effect of hyaluronic acid and stimuli from the culture medium. The test groups contained three xHyA concentrations: 1:100, 1:10, and 1:1 according to the study by Fujioka-Kobayashi et al., 2017 [30].

Test condition preparation was carried out by hydration of each substrate with xHyA using three different dilutions: 1:100, 1:10, and 1:1; standard DPBS was the dilution medium. The controls were hydrated by the regular culture medium. The incubation period for the cell pellets at the substrate–air interface lasted from 7 to 14 days at 37 °C in a humidified atmosphere with 5% CO_2_.

At the end of the incubation period, the collagen substrates were removed from the cell culture insert with sterile forceps and the cells were then carefully scraped from the surface of the collagen substrates with a cell scraper and transferred to a reaction tube. In addition, the collagen substrates were washed with 1 mL of DPBS, which was transferred to the same reaction vessel. Total RNA was processed using the NucleoSpin^®^RNA kit (ThermoFisher Scientific GmbH, Schwerte, Germany) according to the manufacturer’s instructions. The purified total RNA was eluted from the column with 60 µL of RNAse-free water (11,000 × *g*, 60 s, RT) and stored at −80 °C. The cDNA synthesis was performed using random hexamer primers and the First Strand cDNA Synthesis Kit (ThermoFisher Scientific GmbH, Schwerte, Germany) according to the manufacturer’s instructions.

Quantitative PCR (qPCR) determined the relative expression of a target gene in relation to a reference gene. In this study, the following osteogenic markers were investigated: Runx2, BGLAP, IBSP, CX43, and POSTN. GAPDH served as the reference gene (“housekeeping gene”). The primer design (Table 1) was produced using the NCBI/Primerblast website or taken from a database (the webpage is non-active meanwhile). The efficiency and validity of the primers used were previously checked using positive controls. qPCR analysis (total volume of 10 µL per reaction) was performed using the GoTaq^®^ G2 Hot Start Master Mix (Promega, Walldorf, Germany) and 10 μM primers according to the manufacturer’s instructions. The StepOne Plus Real-Time PCR System (ThermoFisher Scientific GmbH, Schwerte, Germany) was used for qPCR analysis, and the relative target gene expression level was determined in relation to GAPDH using the 2^−ΔΔCT^ method.

The transcription of genes encoding for RUNX2, BGLAP, IBSP, and Connexin43 (Cx43) were analyzed by RT-qPCR using SYBR green method^®^.

Multiple comparisons performed using one-way analysis of variance (ANOVA) used ΔΔCT scores calculated in relation to NC and PC after adjusting the expression levels of the targets to housekeeping GAPDH, respectively. Statistical analysis was completed using SPSS 23.0 (USA). Values of *p* < 0.05 were considered significant. To show the relative expression difference in the target genes between the collagen membranes and the two time points, the 2^−ΔΔCT^ values were visualized in bar charts using Microsoft^®^ Excel (version 16.35).

An H&E and a toluidine blue staining were performed to prove that the Saos-2 cells could be cultivated in the air-lift model over a period of 14 days. As an example, the substrate B was used here, and the experimental setup was identical to the main experiments. The culture medium was then replaced by 4% paraformaldehyde so that the cells were not washed off. The fixation lasted for 1 h, and thereafter, the specimens were washed three times with DPBS. The collagen membranes were paraffin-embedded and cut in different dimensions. This was followed by a classical deparaffinization and staining the sections with H&E and toluidine blue.

## 3. Results

After 7 days, the cells on the substrate A hydrogenated with xHyA (concentration 1:1) showed a slightly increased Runx2 expression. In comparison, no change in the expression profile was observed for the other target genes examined (Figure 3A). All target genes showed a strongly increased expression pattern after 14 days of cultivation on the xHyA-treated substrate A (Figure 3C). Runx2 showed an up to 120-fold increase in expression (xHyA 1:10). In comparison, the expression of IBSP increased 8-fold under xHyA 1:10 hydrogenation of the same substrate material. The comparison of the stimulation conditions (xHyA 1:100 vs. xHyA 1:10 vs. xHyA 1:1) showed a heterogeneous expression profile of the investigated target genes. No correlation between the selected xHyA concentration and the upregulation of the investigated target genes could be detected. However, multiple expression was observed under the xHyA addition for all target genes compared to the positive control (differentiation medium without xHyA). For example, the strongest Runx2 and IBSP expression was observed with xHyA 1:10, whereas the strongest BGLAP expression was detected with xHyA 1:1 and of Cx43 with xHyA 1:100.

Compared to the results from the substrate A, clear differences in the expression profiles of the investigated target genes were observed during cultivation on the substrate B, both soaked with xHyA. After 7 days, only xHyA 1:100 showed a 7.5-fold increase in BGLAP expression (Figure 3B). A similar picture emerged after 14 days. Again, strongly increased expression could only be detected with BGLAP (approximately 22-fold), this time, however, with xHyA 1:1 (Figure 3D). Compared to 7 days, up to 4-fold increased expression for the negative control over the positive control was observed for the other target genes examined (Runx2, IBSP, and Cx43) after 14 days. Similar to the data obtained in cells from the substrate A, no correlation between the selected culture conditions (xHyA 1:100 vs. xHyA 1:10 vs. xHyA 1:1) and the expression profile of the investigated target genes could be observed in cells from the substrate B.

### Statistical Analysis

The total expression of the other targets showed a substrate- and time-dependent profile. Delta-delta CT related to PC showed *p* = 0.0001 for culture duration, *p* = 0.028 for the substrate A or B, and *p* = 0.022 for culture and group. All other parameters were statistically nonsignificant. The t-test performed for the unpaired samples for the two collagen substrates (A and B) showed a *p*-value of 0.032 for the substrate A hydrated with 1:100 xHyA only.

Under the xHyA treatment of the substrates A and B, the estimated overall gene expression level in cells seeded on the substrate A grew continuously from day 7 to day 14 while cells seeded on the substrate B revealed a constant expression rate within the 14-day culture period (Figure 4).

The light microscopic images showed pale blue-colored, clearly demarcated collagen substrate B (Figure 5). The cells with cytoplasm, some of which were multilayered, were densely packed all around and had both a spheroidal and a flat outstretched shape. The vitality of the cells after two weeks was evident from the dark blue-colored intact cell nuclei. The H-E staining demonstrated dark pink-colored cells with darker nuclei, clearly arguing for maintained cell activity in cells adhered to the substrate B.

## 4. Discussion

The present study aimed to both prove that collagen materials used as permeable substrates sustain a sufficient nutrition level of osteoblast-like cells and evaluate the impact of cross-linked heavy chain hyaluronic acid (xHyA) on osteogenic differentiation. The continuous transcription of encoded osteogenic proteins during the culture period was assessed by RT-qPCR at two time points. The synergistic efficacy of the combination on cell differentiation and the gene expression profile was addressed by pre-treatment of the two substrates with xHyA.

According to the expression profile for the targeted transcripts, the SAOS-2 cells survived and showed osteogenic differentiation when maintained during the 14 days of the experiment under the air-lift conditions. The supply of the organoid-like multilayer cell arrangement by nutrients via diffusion across both types of cross-linked collagen substrates was apparently sufficient irrespective of the xHyA pre-treatment. This observation is in line with the results from a previous study, which used a similar setup but in combination with different substrates derived from other membrane materials and under xHyA exclusion [13]. The current results suggest that the xenogeneic cross-linked collagen materials clinically used as tissue barrier membranes did not interrupt the cell nutrition in vitro, indicating that neither one acted as a dense diffusion barrier.

Recent research has used the air-lift model to construct multilayer tissue conglomerates in vitro to study chondrocytes, tracheal epithelial cells, or keratinocytes [31,32,33]. Although rather uncommon in the field of bone cell research, earlier studies confirmed the applicability of the air-lift principle to study primary bone and bone-like cell behavior, looking at an organoid culture arrangement [14,34].

As the preliminary experiment revealed a positive Alizarin red reaction (data not shown), we decided to limit the choice of osteogenic stimuli to ascorbic acid and β-glycerophosphate for this study. Calcium forms an Alizarin Red S-calcium complex in a chelation process, and the end product is birefringent. This method is widely used to monitoring the calcification grade in osteogenic differentiation studies [35]. The effect of dexamethasone on osteoblast differentiation and mineralization was controversial, depending on the cell maturation and cell density level [36,37]. Some authors have reported that dexamethasone was not essential for mineralization in culture and long-term use even decreased bone formation [38,39].

The cross-linked HMWHA (xHyA) was considered for this experiment in accordance with preclinical study observations [29,40]. The rationale for choosing the cross-linked HA regards the clinical indication for the collagen membranes serving as substrates, which is bone regeneration. As bone regeneration is a long-lasting regeneration process (between several weeks to several months), we prefer the presence of a medium cross-linked HA in the wound with a prolonged degradation pattern of several weeks. The impact of hyaluronic acid on the expression of osteogenic markers in osteoblast-like cells was assessed after 7 and 14 days of permanent contact between the cell pellet and the xHyA-treated collagen substrate. The increase in the transcription rate from day 7 to day 14 for the target genes Runx2, BGLAP, IBSP, and Cx43 confirmed the continuous osteogenic differentiation of the SAOS-2 cells in this experiment. The results revealed a statistically significant (*p* = 0.0001) difference in the expression rates between both time points. With a consistent increase in gene transcription, the hyaluronic acid effect appeared extensively unfolded after two weeks. Another in vitro study showed the positive effect of xHyA derivatives on improved osteogenesis and mineralization [28].

Moreover, in the presence of xHyA at a 1:10 dilution, a significantly greater expression rate favored the substrate A vs. the expression level obtained from the substrate B after two weeks (*p* = 0.028). Cells seeded on the first one treated with 1:10 xHyA increased the rate of Runx2 expression from 5- to 120-fold during the second half of the culture period. Under this condition, other targeted genes showed similar dynamic expression patterns. A similar xHyA concentration in a 1:10 dilution positively affected osteogenic differentiation in a study by Fujioka-Kobayashi et al. after 7 days (2017). The gap of 7 days in the expression rate between both studies possibly relates to the difference in the experimental setup but probably also reflects differences in the xHyA formulation. In contrast to the air-lift setup of our study, the previous study used the traditional design of seeding the cells on plastic, covering with a layer of xHyA, and then floating the cell layer with a medium in a 24-well plate.

The significantly different gene expression rates in cells sampled from the two collagen substrates after 14 days indicate that the cell response and the degree of collagen cross-linking may be closely associated. According to the morphometrically and histologically documented degradation dynamic for the two basic materials, the native porcine pericardium membrane (NPCM, substrate A) was considered a less cross-linked device compared to RCCM (substrate B) [41,42]. Several groups claimed from their studies that the GTR membrane materials per se differentially affect cell proliferation and differentiation in the process of periodontal tissue regeneration [43,44,45,46]. We can speculate that the loosely arranged collagen structure of the NPPM membrane was more efficiently saturated by the viscous xHyA gel than the very condensed RCCM material [47]. In saying this, we conclude that the NPPM became a kind of “container” device for xHyA, continuously delivering it to the cells seeded upon. A similar observation was made by Eliezer et al. from their experiment using the same material combination in a subcutaneous diabetic rat model [40].

HA has been recognized and extensively investigated as a potent biologic, sharing angiogenetic and osteoinductive properties [23,24,48]. The cross-linked hyaluronic acid as a heavy chain glycosaminoglycan has been subjected to in vitro, pre-clinical, and clinical studies during last decade. Asparuhova et al. (2018) succeeded in demonstrating a positive effect of hyaluronic acid on different cell types after a short cultivation period in vitro (24 and 48 h) [49]. The most recent data from the same group indicated that the addition of xHyA in vitro stimulated cell proliferation rather than supporting the differentiation behavior in osteoblast progenitors. The results emphasized that at the level of growth factors, more pronounced effects were observed with and under xHyA stimulation than activated by differentiation factors in the ST2 and MC3T3-E1 cells. However, unlike the present experiment, the cell response was analyzed after 96 h at the latest; thus, the results must be interpreted with caution, particularly considering the study duration component and the cell’s origin [27]. In this experiment, the late osteogenic marker expression appeared inhibited in the presence of HA in both non-cross-linked and cross-linked HMWHA formulations. In contrast, our results disclose significantly increased expression rates of late osteogenic markers such as BGLAP and Cx43, especially on a loosely cross-linked collagen substrate treated with HMWHA (xHyA) (Figure 3). In addition to these particular findings, our experiment confirmed the impact of xHyA on the transcription output in cells challenged by the air-lift arrangement while a recent experiment on human osteoblast cells just assessed the cell viability under regular 10-day culture conditions [50].

This study did not look at the protein expression level of the cells because of the experimental arrangement. The expression of proteins in the medium through the collagen layer is unlikely to become measurable, therefore no attempts were made to sample the media and to assess the corresponding protein levels. The expression control by IHC using antibodies within the organoid culture was omitted owing to the limited number of samples available. These facts represent study limitations. A further limitation relates to the single cell line used, resulting in the data being applicable to the SaOS-2 cells only. Any extrapolation of the results to other cell lines or in vivo outcomes is not supported.

## 5. Conclusions

Within the limitations of this study, we conclude that sufficient nourishing of cells via diffusion was confirmed. Pre-treatment of collagen substrates with xHyA had an impact on expression rates, favoring the native porcine pericardium membrane (NPPM) vs. RCCM at 7 and 14 days. The air-lift model using NPPM or RCCM effectively supported the gene transcribing activity of SAOS-2 cells at both time periods. Further studies with a high number of replicates are necessary to validate the statistics. To improve our understanding of the xHyA function, an extension of targeted genes towards, e.g., growth factor expression, appears rational for future settings.

## Figures and Tables

**Figure 1 materials-15-06528-f001:**
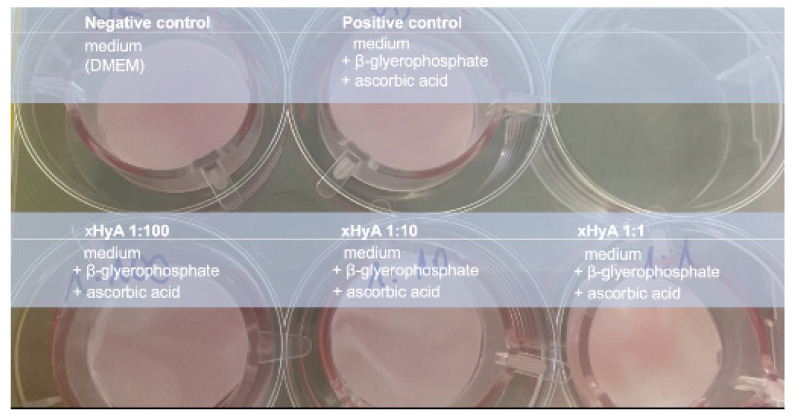
Test conditions disclosed for a single well with the installed insert on a 6-well plate.

**Figure 2 materials-15-06528-f002:**
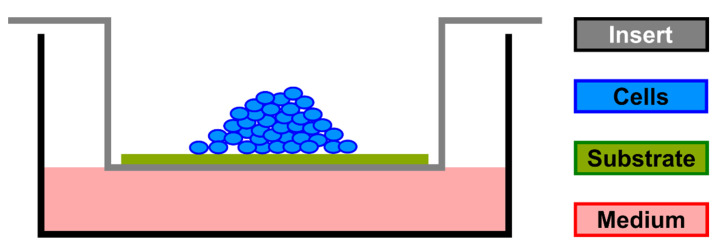
Schematic air-lift model *Substrate A (NPCM +/− xHyA); B (RCCM +/− xHyA).

**Figure 3 materials-15-06528-f003:**
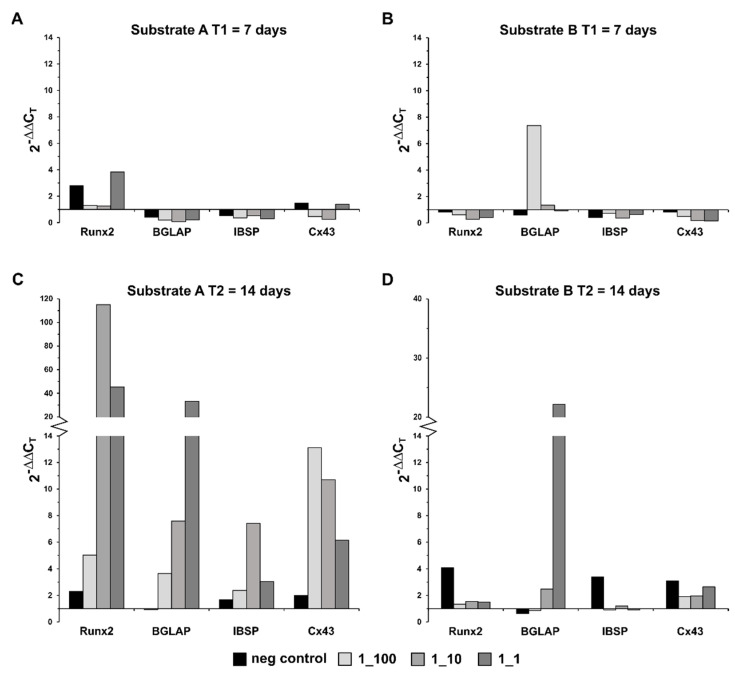
Relative expression differences of the target genes to the housekeeping gene GAPDH compared to the positive control in cells from the substrate A (NPCM, **left**) and substrate B (RCCM, **right**) after 7 and 14 days. (**A**) xHyA in 1_1 dilution used on substrate A enhanced only Runx2 expression by factor 4 at T1, (**B**) on the substrate B only BGLAP displayed enhanced expression rate under xHyA at 1_100 dilution at T1, (**C**) xHyA dilutions of 1_10 and 1_1 on substrate A upraised the expression of Runx2 and BGLAP to 120 and 44 fold, respectively, while highest level of Cx43 expression was measured at 1_100 dilution of xHyA, (**D**) only BGLAP showed increased expression on the substrate B treated with 1_1 xHyA.

**Figure 4 materials-15-06528-f004:**
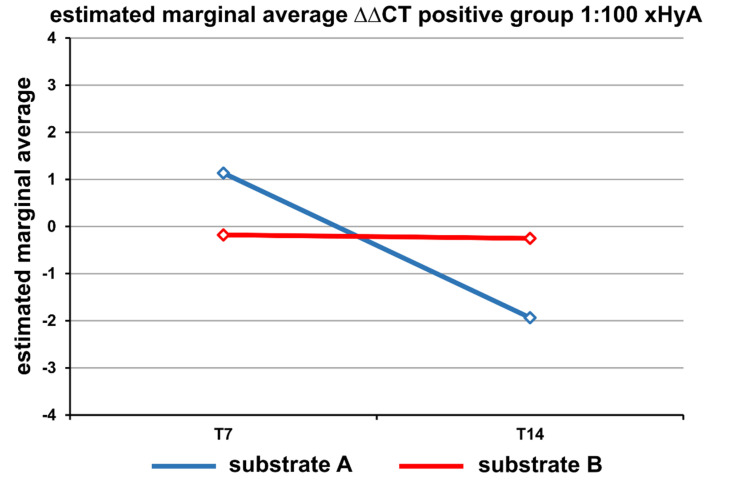
The statistically significant difference in the −ΔΔCT scores between the two substrates.

**Figure 5 materials-15-06528-f005:**
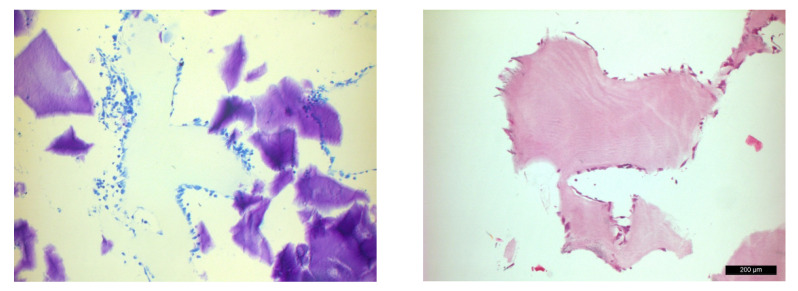
Toluidine blue staining of SAOS-2 cells adhered to the substrate B after 14 days (**left**) and hemalum-eosin staining of SAOS-2 cells on the same substrate B (**right**).

**Table 1 materials-15-06528-t001:** Summary of pPCR primers.

Name	Primer	Sequence (5′ to 3′)
Cx43	forward	CCT TCT TGC TGA TCC AGT GGT AC
	reverse	ACC AAG GAC ACC AVV AGC AT
BGLAP	forward	TTC TTT CCT CTT CCC CTT G
	reverse	CCT CTT CTG GAG TTT ATT TGG
IBSP	forward	GGA GAC TTC AAA TGA AGG AG
	reverse	CAG AAA GTG TGG TAT TCT CAG
Runx2	forward	CCA ACC CAC GAA TGC ACT ATC
	reverse	TAG TGA GTG GTG GCG GAC ATA C
GAPDH	forward	TGC ACC ACC AAC TGC TTA GC
	reverse	GGC ATG GAC TGT GGT CAT GAG

## Data Availability

Not applicable.

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
