# Peer review of "Impact of Cross-Linked Hyaluronic Acid on Osteogenic Differentiation of SAOS-2 Cells in an Air-Lift Model"

_materials, 2022, doi:10.3390/ma15196528_

Round 1
Reviewer 1 Report
The article deals with the impact of cross-linked hyaluronic acid on osteoblast-like cells. Here follows some comments:
I ask to explain better the air-lift model, I don't know what it is, I recommend a wider explanation and to improve the scientific background in the introduction. What is new compared to ref 14? Explain better the innovation in this paper. In the statistic part indicate the level of significance. What does it mean that the Alizarin red reaction is positive?
Minor comments:
Check the font after keywords, the font of Figure 1, Figure 2, figure 4. In the legend of Figure 4 add e to membran.
Author Response
Dear Reviewer,
thank you for the comments and concerns you shared with us. We extended some informations as recommended. The changes in the manuscript are highlighted by the correction mode in WORD document.
- "While previous experiment used metal grids to lift up the cell pellet above the medium level, today disposable inserts for 6-well plates are available for the purpose [13]. Thus, placing the cell pellet either on top of the insert's membrane or on an additional substrate adopted to in size, the nutrition progresses only via diffusion, as the cells are detached from direct contact with the medium."
- "The previous experiment confirmed that the three questioned substrates were capable of keeping the permeability level upright to nourish the cells sufficiently assassed by their transcription rates. In this study, one different substrate was included and using the same osteogenic additives as in the past experiment, we inclued the cross-linked hyaluronic acid treatment to the substrates to evaluate the potential impact on the transcription rate of the bone-like cells."
- The striking outcome in our study and the previous in vitro study by Asparuhova et al. are now discussed in detail. Also additional studies are included and discussed in light of our findings.
"In this experiment the late osteogeic marker expression appeared inhibited in presence of HA in both, non cross-linked and cross-linked HMWHA formulations [50]. In contrast, our results disclosed significantly increased expression rates of late osteogenic markers as BGLAP and Cx43 especially on a loosely cross-linked collagen substrate treated with HMWHA (xHyA) (Fig. 3). Besides these particular findings our experiment confirmed the impact of xHyA on the transcription output in cells challenged by the air-lift arrangement while recent experiment on human osteoblast cells just assessed the cell viability [51]."
- The concern regarding the Alizarin red reaction is now addressed in the Discussion section as followig: "The alizarin red method assesses the calcification progress in a cell culture. Calcium forms an Alizarin Red S-calcium complex in a chelation process, and the end product is birefringent. The method is widely used to monitoring the calcification degree in osteogenic differentiation studies (34)."
Thank you for the advice to adjust the format of the font throughout the manuscript, it has been adopted.

Reviewer 2 Report
This is an interesting study about impact of cross-linked hyaluronic acid on oteogenic differentiation of SAOS-2 Cells. I recommend it for publication after the issue below are addressed.
1. Line 18, 'synovial fluid'. One recent study (Cells 9 (7), 1606) should be included to support such a claim.
2. As a material journal, some information about cross-linked HA is missing, such as the molecular weight of HA, the cross-linking degree, and chemistry.
3. Why was cross-linked HA chosen for this study, but not free HA polymers?
4. Was the cross-linked HA degraded in the experimental conditions?
5. Technical issues. Keep the font the same through the ms. The resolution and quality of the figures must be improved.
Author Response
Dear Reviewer,
thank you for the comments and recommendations. We adopted the manuscript accordingly. The changes you find below listed point by point. The changes in the manuscript are highlighted by the correction mode in WORD document.
1. According to your recommendation the following citation is included:
"The Role of Hyaluronic Acid in Cartilage Boundary Lubrication. W. Lin, Z. Liu, N. Kampf and J. Klein Cells 2020 Vol. 9 Issue 7".
2.-4. The concerns regarding the HA formulation have been addressed in the M&M (1) and discussion (2) sections:
(1) "High molecular weight hyaluronic acid formulation with ca. 1 MegaDalton (1000 kDa) was chosen in cross-linked condition. The degree of cross-linking corresponded to the medium cross-linking rate with a degradation time of approximately 3-4 weeks according to the manufacturer (Regedent Ag, Switzerland). The cross-linking agent was: BDDE (1.4-butanediol diglycidyl-ether)."
(2) "The cross-linked HMWHA (xHyA) was considered for this experiment in accordance to preclinical study observations [29, 40]. The rationale for choosing the cross-linked HA regards to the clinical indication for the collagen membranes serving as substrates, which is bone regeneration. As bone regeneration is a long-lasting regeneration process (between several weeks to several months) we prefer the presence of a medium cross-linked HA in the wound with a prolonged degradation pattern of several weeks. "
5. Thank you for the advice that the figures included had too poor quality. All images have been exchange by figures with higher resolution and higher quality.

Round 2
Reviewer 2 Report
accepted in the current form